# From PROTAC to TPD: Advances and Opportunities in Targeted Protein Degradation

**DOI:** 10.3390/ph17010100

**Published:** 2024-01-11

**Authors:** Siqi Wang, Fuchu He, Chunyan Tian, Aihua Sun

**Affiliations:** 1State Key Laboratory of Medical Proteomics, National Center for Protein Sciences (Beijing), Beijing Proteome Research Center, Beijing Institute of Lifeomics, Beijing 102206, China; wangsq0522@163.com (S.W.); hefc@bmi.ac.cn (F.H.); 2Research Unit of Proteomics Dirven Cancer Precision Medicine, Chinese Academy of Medical Sciences, Beijing 100050, China

**Keywords:** PROTAC, TPD, targeted protein degradation, protein-hydrolysis-targeted chimeras, ubiquitin–proteasome system

## Abstract

PROTAC is a rapidly developing engineering technology for targeted protein degradation using the ubiquitin–proteasome system, which has promising applications for inflammatory diseases, neurodegenerative diseases, and malignant tumors. This paper gives a brief overview of the development and design principles of PROTAC, with a special focus on PROTAC-based explorations in recent years aimed at achieving controlled protein degradation and improving the bioavailability of PROTAC, as well as TPD technologies that use other pathways such as autophagy and lysosomes to achieve targeted protein degradation.

## 1. Introduction

From salicylic acid derived from willow to imatinib, the first targeted small-molecule drug against BCR-ABL, the development of small-molecule inhibitors has undergone a brilliant history from empirical science to rational design. However, their occupancy-driven mechanism dictates that small-molecule inhibitors have difficulty effectively inhibiting the activity of pathogenic proteins with wide and shallow pockets or smooth surfaces and are prone to drug resistance. In addition to the recent boom in biological therapies, targeted protein degradation (TPD) has also gained much attention from researchers, offering new opportunities to target proteins that are traditionally defined as “non-druggable”.

In this area, the protein-hydrolysis-targeting chimera (PROTAC) has received the most attention and has been the most extensively studied and advanced clinically. PROTAC consists of a warhead that binds to the protein of interest (POI), an E3 ubiquitin ligase ligand that recruits E3 ubiquitin ligase, and a linker. After forming a POI–PROTAC–E3 ubiquitin ligase ternary complex, PROTAC induces the ubiquitination of the POI and its subsequent degradation by the 26S proteasome. This event-driven mechanism of action makes it completely different from traditional small-molecule inhibitors.

This paper gives a brief overview of the development and common design principles of PROTAC, with a focus on summarizing new explorations based on PROTAC, and TPD strategies benefiting patients via other pathways.

## 2. The Development of PROTAC

### 2.1. PROTAC Theoretical Basis

PROTAC degrades target proteins via the ubiquitin–proteasome system (UPS) (Figure 1). UPS is involved in the regulation of numerous biological processes in eukaryotic cells, and the ubiquitination of proteins is one of the most important determinants of cell fate and function. A misfunctioning UPS may lead to the development of congenital defects and many major diseases.

Ubiquitin (Ub) is a 76-amino acid protein with a highly conserved sequence, which can be specifically expressed by different ubiquitin-encoding genes under different physiological conditions. Ubiquitination involves three main steps named activation, conjugation, and ligation, performed by ubiquitin-activating enzymes (E1s), ubiquitin-conjugating enzymes (E2s), and ubiquitin ligases (E3s), respectively. Proteins can be mono- or polyubiquitinated, and the formation of ubiquitin chains of different lengths and topologies often confers different fates on proteins, where a specific length of K48 ubiquitination modification will induce protein degradation via the 26S proteasome [1,2,3].

PROTAC accelerates the degradation of the intrinsic substrate by forming a ternary complex or degrades proteins that are not otherwise degraded by the proteasome in their physiological state. Unlike traditional small-molecule inhibitors, this mechanism of action allows PROTAC to act without occupying the active site of the protein, which is particularly suitable for targeting pathogenic proteins with multiple active sites, shallow binding pockets, or smooth surfaces lacking binding pockets [4] Theoretically, due to the direct reduction in POI levels, PROTAC is less likely to cause mutational resistance and even degrade wild-type/mutant POIs simultaneously.

In recent years, the concept of molecular glue has often been mentioned in conjunction with PROTAC, but differences in the mechanisms of action make it impossible to generalize between these molecules (Figure 2). Cyclosporine A was first isolated from soil samples by scientists at Sandoz Pharmaceuticals in 1969, but it was not until 1991 that the Schreiber laboratory at Harvard University first described the mechanism of cyclosporine A and FK506 as a natural molecular glue [5]. Two decades later, the identification of the direct target of thalidomide dramatically drove the rise of the concept of molecular glue degradation agent [6]. In the past, thalidomide was thought to be a drug with multiple targets and benefited patients with certain diseases or caused fetal malformations by interacting directly with different proteins. Thalidomide binds to cereblon (CRBN), which is a substrate recognition receptor for Cullin 4 RING (CRL4) E3 ubiquitin ligase and induces the recruitment of nonintrinsic substrate to CRL4^CRBN^ and their subsequent degradation. Thalidomide and its derivatives, now known as immunomodulatory imide drugs (IMiDs), have also become well-established ligands for the recruitment of CRL4^CRBN^ E3 ubiquitin ligases commonly used in PROTAC.

### 2.2. PROTACs Bridge the Gap between Preclinical Research and Clinical Application

In 2001, the concept of PROTAC was first introduced in the Crews lab at Yale University [7]. The first PROTAC, Protac-1, was designed and synthesized to achieve the degradation of methionine aminopeptidase-2 (MetAP-2), which is neither a native substrate of SCFβ-TRCP E3 ubiquitin ligase nor being degraded via the proteasome pathway under physiological condition. It consists of a POI-binding warhead, the antiangiogenic drug Ovalicin, a linker, and an IκBα phosphorylated peptide that recruits the SCFβ-TRCP E3 ubiquitin ligase. This type of peptide-containing PROTAC is now called bioPROTAC, and its application is limited mainly due to the peptide’s susceptibility to hydrolysis.

VHL is another E3 ubiquitin ligase commonly used in PROTAC, and HIF-1α is its main substrate in the physiological state. In 2012, Ciulli A and Crews CM designed and synthesized a small-molecule analog of the HIF-1α recognition peptide [8], which makes the PROTAC based wholly on small molecule structures a reality and opens the door for PROTAC to become an orally bioavailable and thus easy-to-use drug candidate.

Since 2013, pharmaceutical giants such as Merck, Novartis, and Pfizer have laid out their PROTAC pipeline, and companies such as Arvinas and C4 Therapeutics have also been established and focused on PROTAC development. Finally, Arvinas, with ARV-110 for metastatic castration-resistant prostate cancer [9] and ARV-471 for ER^+^/HER2^−^ breast cancer [10], first provided a clinical proof-of-concept for PROTAC in mature tumor targets, demonstrating the ability to degrade target proteins in vivo and providing clinical benefit to patients [4]. In terms of safety, ARV-110 is the first to be tested in a mature tumor target. In terms of safety, ARV-110 and ARV-471 have not been observed to have dose-limiting toxicity in dose-creep clinical trials up to 420 mg/700 mg, respectively. Encouraged by the encouraging result in the Phase I clinical trial, both agents have advanced to Phase Ⅱ and are being explored for a comedication regimen with other anticancer therapies (Table 1).

### 2.3. A Brief Glimpse of Published PCT Patent Application of PROTACs

Although only a limited number of PROTACs have successfully entered clinical trials, a large number of PROTACs targeting different POIs have been patented, as described in an excellent review [15]. Here, we briefly summarized them into four categories: PROTACs targeting nuclear receptors, kinases, epigenetic mechanisms, and misfolded proteins, each of which is exemplified by a patented PROTAC molecule (Figure 3).

### 2.4. PROTAC Targets for Cancer Therapy

#### 2.4.1. PROTAC Targeting AR/ER

Hormonal therapies that block androgen generation or directly inhibit the androgen receptor are major therapies for patients with advanced prostate cancer [16,17]. The aforementioned ARV-110, also known as bavdegalutamide, induced potent AR degradation in prostate cancer cell lines with a DC_50_ of ~1 nM and reduced prostate-specific antigen (PSA) levels greater than or equal to 50% (PSA_50_) in 46% of metastatic castration-resistant prostate cancer (mCRPC) patients with tumors harboring AR T878X and/or H875Y (X = A or S) mutations in its Phase I/II clinical trial [9,18]. The ability to degrade not only wild-type AR but also mutants indicates its potential application as a second-line therapy for patients with primary/acquired resistance to hormonal therapy, especially to AR-targeting agents.

Breast cancer is the second leading cause of death for women worldwide. Estrogen-receptor-positive (ER^+^) patients account for nearly 80% of breast cancers; therefore, endocrine therapies blocking the estrogen–ER axis have remained the standard treatment for these patients for several decades [19,20]. Similar to ARV-110, ARV-471 can degrade both wild-type ER and mutants such as Y547S and D538G, addressing the deficiencies of traditional ER antagonists [10]. Arvinas also evaluated the possibility of ARV-471 combining with the CDK4/6 inhibitor palbociclib. This combination yielded a 131% tumor shrinkage in the preclinical CDX model, supporting the ARV-471–palbociclib combination cohort as a part of the Phase I/II clinical trial in patients with advanced-stage ER^+^HER^−^ breast cancer.

#### 2.4.2. PROTAC Targeting BTK

Bruton’s tyrosine kinase (BTK) is an essential component of multiple signaling pathways that regulate B cell and myeloid cell proliferation, survival, and functions [21]. CRBN-based protein degraders that target BTK have been developed for patients with B-cell malignancies. Both NX-2127 and NX-5948 are capable of degrading wild-type and mutant BTK, while NX5948 does not affect IkZF1/3, thus precluding immunomodulatory effects associated with IMiD CRBN-targeting moieties [11,12,22,23].

#### 2.4.3. Other Targets Involved in Carcinogenesis

CFT-8634 is a BiDAC^TM^ degrader targeting BRD9 for the treatment of BRD9-dependent cancers, including synovial sarcoma and SMARCB1-deleted cancers. Bromodomain-containing protein 9(BRD9), a component of the noncanonical BAF complex responsible for chromatin remodeling, has been recognized as an appealing therapeutic target in hematological malignancies [24,25,26].

The epidermal growth factor receptor (EGFR) regulates cell proliferation and multiple signal pathways. The alteration of EGFR signaling is associated with tumor growth, angiogenesis, invasion, and metastasis [27]. Tyrosine kinase inhibitors (TKIs) targeting EGFR have been applied in cancer treatment and have shown considerable efficacy in patients with non-small cell lung cancer, breast cancer, glioma, etc. However, plenty of patients suffer from intrinsic and acquired resistance mediated by EGFR, making PROTAC a promising approach because of its capability to degrade wild-type and multiple mutants simultaneously [28,29].

Moreover, other essential proteins involved in tumor initiation, proliferation, metastasis, and apoptosis could be potential targets for PROTAC degradation, especially those with multiple mutants or difficult to target with small-molecule inhibitors.

## 3. PROTAC Design Strategy

### 3.1. POI and Warhead

The first PROTACs to enter the clinical phase targeted classical, clinically proven targets, answering the questions of in vivo safety and efficacy and establishing PROTAC as an effective treatment for solid tumors. However, the real exciting prospect of PROTAC is to rewrite the history of some of the “hard to target” or “non-druggable” proteins, providing a targeting strategy that is completely different from that of traditional small-molecule inhibitors.

Unfortunately, no investigator has yet summarized a “gold standard” for PROTAC targets similar to the rule of five proposed by Linpinski. The ideal PROTAC target is generally considered to (1) undergo pathogenic gain-of-function alterations, such as overexpression, mutation, and altered location; (2) have a pocket where a warhead can bind; (3) have a surface site that can be ubiquitinated by E3 ubiquitin ligase; (4) preferably have a nonrigid structural region that can enter the barrel cavity of the proteasome [4,30].

PROTACs currently under investigation tend to use targeted medicines that have already been used in the clinical treatment or well-studied inhibitors that have entered clinical trials as a warhead. These molecules are usually supported by extensive in vitro/in vivo experiments, which can reduce the risk of the warhead not being able to bind effectively to the POI.

It is worth mentioning that a high affinity or covalent binding between the POI and the warhead is not required for a PROTAC. A study based on the multikinase inhibitor foretinib suggested that PROTAC with foretinib as the warhead only degraded a small fraction of the kinases present bound to foretinib, and the final efficiency of kinase degradation was independent of the affinity between foretinib and the target protein, instead of depending on the efficiency of the formation of POI–PROTAC–E3 ubiquitin ligase ternary [31]. In addition, the high affinity or covalent binding between the POI and the warhead may make it difficult to free PROTAC from the ternary complex to achieve long-lasting protein degradation, and it tends to act through an occupancy-driven rather than an event-driven mechanism, losing the significance of the PROTAC application.

### 3.2. Linker

The primary issue to consider in linker design is the selection of suitable sites in the warhead and the E3 ubiquitin ligase ligand. The ideal linker site allows the linker to approach from the ligand-binding pocket or solvent-exposed region, trying to avoid any effect on the binding between the small molecule and its target protein. Considering the accessibility and yield of the reaction, researchers usually look for linker sites from the active atoms of the ligand, such as carboxyl and amino groups [32]. For some larger ligands, the linker can be attached while removing some irrelevant groups without affecting POI targeting or E3 ubiquitin ligase recruitment [33]. For small molecules with multiple potential linker sites, it is important to consider whether some of the groups in the solvent-exposed region are also required for binding between the small molecule and the protein [34,35], which often requires experimental verification.

No universally accepted principle of linker design ensures the degradation of target proteins. Finding a suitable linker can require a lot of experiments and may be influenced by the established method in different laboratories. PEG and alkyl chains are the most common linker motifs, as researchers can easily synthesize these chains of different lengths to compare their degradation efficiency on the target protein. Click chemistry is often used for the synthesis of PROTAC molecules due to its high efficiency, rapidity, and mild conditions, so triazole structures are also a common choice of linkers [32]. Some researchers have also focused on the development of linkers. Li H and his coworkers designed and synthesized a dual-targeted PROTAC using the natural star-shaped structure of amino acids to achieve the simultaneous targeting of EGFR and PARP, which is a novel attempt [36] (Figure 4).

The difficulty of linker design is that it is impossible to precisely predict in advance the linker length or specific conformation required to achieve the ubiquitination of the POI by the recruited E3 ubiquitin ligase. A linker that is too short tends to form a POI–linker or E3 ubiquitin ligase–linker binary complex rather than a POI–linker–E3 ubiquitin ligase ternary complex. On the other hand, too-long linkers also reduce the stability of the formed ternary complexes and affect the ability of PROTAC to penetrate cell membranes.

### 3.3. E3 Ubiquitin Ligase

There are more than 600 E3 ubiquitin ligases in the human body. However, among more than a dozen PROTACs currently entering clinical trials, except DT2216, a BCL-xL degrader that circumvents platelet toxicity by recruiting VHL-targeted [37], all PROTACs achieve targeted degradation of the POI by recruiting CRBN.

Finding a suitable E3 ubiquitin ligase ligand has become a major difficulty limiting the application of E3 ubiquitin ligases in PROTAC. Before IMiDs were widely used for CRBN recruitment, investigators had developed PROTACs that recruited mouse double minute 2 homolog (MDM2) and cellular inhibitor of apoptosis protein 1 (cIAP1) E3 ubiquitin ligases [32]. MDM2 acts as an E3 ubiquitin ligase to mediate the degradation of p53 via the proteasome pathway [38], and its ligand nutlin-3a and idasanutlin have been used in PROTACs to recruit MDM2 to degrade AR and BRD4, respectively [39,40]. However, these PROTACs have not been further advanced due to a low degradation efficiency, large molecule weight, and difficulties in synthesis. cIAP1 is a highly conserved endogenous antiapoptotic factor that is overexpressed in a variety of cancers. Naito M found that the methyl bestatin compound (MeBS) could activate the E3 ubiquitin ligase activity of cIAP1 [41] and achieve the degradation of cIAP1 through self-ubiquitination. Off-target cIAP1 autoubiquitination degradation was present in a PROTAC developed based on MeBS [42]. Multiple PROTACs have been designed and developed for different targets [43,44]; however, even with the development of new cIAP1 ligands, this self-ubiquitination has always failed to be avoided, which largely limited the application of cIAP1 as an E3 ubiquitin ligase in PROTACs.

In addition, researchers have paid attention to E3 ubiquitin ligases that are highly expressed in tumors, hoping that the tissue specificity of these E3 ligases will enhance the antitumor activity of PROTACs while reducing toxicity to other tissues. At the same time, the high expression of E3 ubiquitin ligases in such tumors often suggests that the tumor chose to depend on this E3 ubiquitin ligase during its evolution, which to some extent reduces the possibility of tumor cells acquiring drug resistance through E3 ubiquitin ligase deletion mutations or expression downregulation, while a similar resistance has been seen in PROTACs recruiting CRBN and VHL [4]. However, it has also been noted that such E3 ubiquitin ligases tend to be associated with the cell cycle and that the immune and hematopoietic systems, which are also dependent on rapid cell division, may also be affected [4,45]. The therapeutic window and hematopoietic toxicity need special attention during the development process of this kind of PROTAC.

## 4. New Explorations Based on PROTACs

### 4.1. PhotoPROTAC

Controlling the production or release of active small molecules by light has become a relatively mature idea for precision therapy, such as photodynamic therapy (PDT), which treats specific skin diseases and malignancies by activating a photosensitizer that is toxic at the lesion site with specific wavelengths of light [46]. Similarly, researchers hope to reduce drug toxicity by activating PROTACs at specific sites to make them capable of degrading target proteins while the PROTAC remain inactive in other tissues to achieve “systemic drug delivery and precise degradation”.

The photoswitchable PROTAC achieves a reversible control of protein degradation by introducing light-sensitive conformationally altered groups (e.g., azobenzene) in the linker or E3 ubiquitin ligase ligand (Figure 5). For example, Crews CM and Carreira EM changed the PEG Linker of ARV-771 to an azobenzene linker. Under a 415 nm light irradiation, azobenzene was in the *trans* conformation, the distance between the two amide bonds at both ends of the linker was 11 Å, which was consistent with the original ARV-771 linker, and the molecular was activated to have the ability to induce POI degradation; while at the 530 nm irradiation, azobenzene was in the *cis* conformation, and the distance between the two amide bonds was shortened to 8 Å. Thus, the POI could not be ubiquitinated normally and the PROTAC was inactivated [47]. This approach has also been applied in the development of bifunctional molecules targeting FKBP12, bromodomains, and multiple kinases. The tissue penetration of activating lighting seems to be one of the main constraints of the clinical application of photoswitchable PROTACs; however, recent development in implantable localized irradiation and optofluidic drug delivery system will certainly help in expanding these PROTACs’ applicability as a promising photomedicine [48].

Another idea of PhotoPROTAC is to introduce a light-unstable caging group, such as 6-nitroveratryloxycarbonyl (NVOC), to one end of the E3 ubiquitin ligase ligand (Figure 6). Photocaged PROTAC is inactivated in the absence of light and cannot recruit E3 ubiquitin ligase properly; after light activation, the NVOC group is removed, and the PROTAC can form the ternary complex normally to achieve a targeted degradation of the POI [49].

### 4.2. CLIPTAC

PROTAC contains two covalently linked ligands, which are usually large in molecular weight and have a relatively poor solubility, drug metabolic properties, and bioavailability. Click chemistry, pioneered by the Sharpless lab, has been used to describe a series of chemical reactions with the properties of mild reaction conditions, high product yields, and high selectivity. It is now the theoretical basis for covalent modifications and self-assembly under physiological conditions [50,51]. Heightman TD proposed the concept of CLIPTAC in 2016 [52], in which a PROTAC was divided into two precursor drugs, POI–half-linker and half-linker–E3 ubiquitin ligase ligand, which were administered separately and then self-assembled intracellularly to form a complete PROTAC structure and induce POI degradation (Figure 7). Although this method faces the problem of intracellular reaction efficiency, it is still an interesting attempt to improve the bioavailability of PROTACs.

### 4.3. Oligonucleotide-Based PROTAC

The abnormal expression or mutation of transcription factors (TFs) and RNA-binding proteins (RBPs) are important triggers of many major diseases. Protein–protein interaction (PPI) is important for TFs and RBPs to function [53,54]. Different from enzyme–substrate or ligand–receptor interaction, PPI-associated proteins have shallower binding pockets and often multiple active sites for nucleotides or proteins to bind [55]. This makes these proteins difficult to target by traditional small-molecule inhibitors and is often considered “non-druggable”. Such characteristics make TFs and RBPs suitable for PROTAC development.

The development of the bromodomain-containing protein 4 (BRD4) PROTAC is a classic work to target TF [56]. BRD4 is a member of the bromodomain and extra terminal (BET) family, which consists of epigenetic regulatory proteins and transcriptional regulators with important roles in transcriptional initiation, elongation, regulation, and even DNA damage repair [57]. It has been shown that inhibition of BRD4 using JQ1 can reduce intracellular MYC levels, further leading to a reduced expression of MYC downstream target genes, making the inhibition of transcription-factor auxiliary proteins one of the strategies to inhibit transcription factors [58]. What is particularly exciting is the fact that PROTACs targeting BRD4 can effectively overcome the weakness of small-molecule inhibitors that cannot completely inhibit the activity of BRD4 and further reduce MYC levels [59].

Researchers have also made attempts to design universal TF-PROTACs that directly target transcription factors. Jin J and Wei WY designed DNA oligomer-based PROTACs that directly use NF-κB recognition sequences as POI-targeting warheads, which are linked to VHL ligands via a triazole linker to achieve the targeted degradation of NF-κB [60].

Similarly, Hall J designed an RNA-PROTAC based on RNA oligomers and achieved the intracellular degradation of the RNA-binding protein LIN28 by the 26S proteasome, whose warhead is also an intrinsic recognition sequence for LIN28 [61].

Like other oligonucleotide drugs, TF-PROTACs and RNA-PROTACs face dilemmas such as poor pharmacokinetic properties, off-target toxicity, and immature delivery systems [62]. Compared to the TF-PROTAC, the clinical application of the RNA-PROTAC faces even more difficulties, attributed to the high structural variability and changing conformations of RBPs in order to recognize different RNAs [63]. Oligonucleotide-based PROTACs still have a lot of hurdles to conquer to become clinically applicable medicine.

## 5. Alternative Pathways for TPD

The success of PROTACs has led to the investigation of other bifunctional molecules that degrade target proteins by non-UPS pathways, which usually target intracytoplasmic proteins due to the mechanism of action of UPS, while targeting protein degradation by other pathways may broaden the target to membrane proteins, extracellular proteins, and even nonprotein organelles. This also makes these non-UPS-dependent TPDs potentially superior options in certain diseases where the therapeutic target is a noncytosolic protein or nonprotein organelles.

### 5.1. Autophagy–Lysosome Pathway

Autophagy is a highly conserved cellular degradation mechanism in eukaryotes and is mainly classified into macroautophagy, microautophagy, and chaperone-mediated autophagy (CMA). The main function of autophagy is to degrade intracellular substances to provide nutrients and raw materials for critical cellular life activities in stressful conditions such as nutrient deprivation and growth factor deficiency, and thus has long been considered a nonselective process, whereas recent studies have found that autophagy can selectively remove potentially harmful intracellular substances, such as misfolded proteins or damaged mitochondria, suggesting its function as a cytoprotective system [64].

Macroautophagy is mediated by the autophagosome. The process of canonical macroautophagy can be divided into different stages, including initiation, nucleation, elongation, maturation, and infusion. Normally, the autophagy reaction is mainly regulated by rapamycin complex 1 (mTORC1) and 5′ AMP-activating kinase (AMPK) and requires the cascaded cooperation of various proteins, such as unc-51-like autophagy-activating kinase 1 (ULK1), a key protein kinase initiating autophagy, microtubule-associated protein 1 light chain 3 (LC3s) involved in autophagosome formation, and p62 that function as a selective autophagy receptor [64,65].

Recently, researchers have developed techniques that achieve targeted protein degradation via the autophagy process, three of which are reviewed here (Figure 8).

#### 5.1.1. ATTEC

Lu BX of Fudan University proposed that linking POI with LC3, an important marker protein on the phagosome surface, through bifunctional small molecules might mediate the subsequent autophagic degradation of POIs and screened by small-molecule microarray (SMM) technology to obtain small molecules that could simultaneously bind LC3 and a POI mutant huntingtin (mHTT). In that study, researchers achieved the degradation of mHTT via the autophagy process, while wild-type HTT was almost unaffected [66,67]. They also developed LD-ATTECs with Sudan III or Sudan IV dyes as warheads to achieve the degradation of intracellular lipid droplets via the autophagic pathway and validated them in a mouse model [68].

#### 5.1.2. AUTOTAC

p62 mediates the transfer of ubiquitinated autophagic substrates into autophagic vesicles and is subjected to further degradation [69]. Similar to ATTECs, Kim YK and his co-workers developed a bifunctional AUTOTAC molecule that bound to both POI and p62 zinc finger structural domains and directed the POI into the subsequent autophagy–lysosome pathway for a degradation independent of POI ubiquitination [67,70].

#### 5.1.3. AUTAC

8-nitrocyclic guanosine monophosphate (8-nitro-cGMP) is a second messenger downstream of NO that mediates S-guanosylation modifications of protein and further induces autophagy [71,72]. The AUTAC molecule designed and synthesized in Arimoto’s laboratory consists of a warhead that binds to POIs or organelles, a cGMP-based autophagic degradation tag, and a linker. In addition to degrading intracellular POIs, researchers also designed AUTAC4 which achieved the degradation of damaged mitochondria via the autophagy–lysosome pathway by targeting the translocator protein (TSPO) on the outer mitochondrial membrane, providing a new candidate therapeutic approach to mitochondrial-dysfunction-related diseases [73].

### 5.2. Lysosomal Pathway

Lysosomes are single membrane vesicular organelles, containing a variety of endoplasmic-reticulum-synthesized, mannose-6 phosphate-modified hydrolases, capable of degrading a variety of biological macromolecules such as nucleic acids, proteins, and lipids [74]. Traditionally, lysosomes have been considered the “waste recycling center” of the cell, relatively isolated from other organelles. In recent years, as their study has progressed, researchers have found that lysosomes are key organelles involved in the degradation of biomolecules, intracellular transport of substances, innate and adaptive immune activation, and nutrient sensing, and their dysfunction may lead to metabolic disorders, neurodegenerative diseases, and cancer [74,75,76].

Unlike the autophagic pathway, the endosomal–lysosomal pathway is capable of degrading either cytosolic or extracellular proteins. The internalization of extracellular substances is often dependent on the binding of ligands to receptors, and the corresponding process is called receptor-mediated endocytosis (RME). Ligand–receptor complexes are endocytosed into intracellular vesicles, which subsequently fuse with each other to form early endosomes (EEs), where metabotropic receptors are usually recycled to the plasma membrane, while signaling receptors and their ligands are transported to late endosomes (LEs) for subsequent degradation by lysosomes [65]. Unlike PROTACs, which can only degrade intracellular proteins, TPD based on the lysosomal pathway can theoretically degrade both cytosolic and extracellular proteins, further broadening the range of targets (Figure 9).

#### 5.2.1. CI-M6PR LYTAC

The non-cation-independent mannose-6-phosphate receptor (CI-M6PR) is a classical lysosomal targeting receptor that transports proteins with M6P multimeric modifications at the N-terminal end to the lysosome [77]. The first LYTAC molecule was designed and synthesized by Bertozzi CR [78]. LYTAC uses an M6P polysaccharide as a ligand to recruit CI-M6PR, while the other end binds to the extracellular structural domain of the cytosolic protein or extracellular protein. In that study, the investigators used biotin as the warhead to achieve the degradation of extracellular tool protein NA-647, and cetuximab, a clinical monoclonal antibody for EGFR, as the warhead to degrade EGFR successfully.

#### 5.2.2. IFLD

In addition to M6P/CI-M6PR, other receptors on the cell surface involved in ligand–receptor interactions, such as transferrin receptors [79], folate receptors [80], integrins [81], etc., are also able to deliver fluorophores, medicine, or nanoparticles into the cell via RME. Integrin αvβ3 is overexpressed in a variety of tumor cells, and its recognition motif Arg-Gly-Asp (RGD) is widely used in the development of targeted therapies, but no RGD-based TPD studies are available. Fang Lijing and co-workers at the University of Chinese Academy of Sciences designed an integrin-facilitated lysosomal degradation (IFLD)-based strategy [82], which targeted the POI at one end and the integrin recognition sequence RGD at the other end and enabled the degradation of the extracellular tool protein NAP-650 and the cytosolic protein PD-L1 via the lysosomal pathway.

To sum up, we have described several new explorations based on PROTAC and TPDs that take advantage of autophagy or lysosomal pathways to achieve target proteins or organelles degradation (Figure 10). All these efforts attempt to achieve a precise control of medicine activation, improve the bioavailability of such large molecules, or expand the target spectrum, with which we hope that TPDs could be applied clinically in treating patients.

## 6. Conclusions

Over the past two decades, PROTACs have made the first leap from the laboratory to the clinic, completing the theoretical validation of TPD as a therapeutic tool. However, many questions remain to be explored and solved.

Expanding the POI spectrum is a promising orientation for TPD development. The POI of traditional PROTACs is limited to cytosolic proteins. However, many proteins that significantly contribute to cancer development and metastasis are located on cell membrane and act as receptors. Researchers have made an effort to expand the POI spectrum to membrane-bound and circulating proteins, and even further, lipids and organelles, by taking advantage of autophagy and lysosomal pathways.

Another obstacle keeping PROTACs from clinical application is the poor bioavailability of most PROTAC molecules. The structure of three moieties determines that PROTACs tend to have large molecular weights, making it difficult to penetrate cell membranes. Researchers attempt to utilize click chemistry, administrating precursors of fully functional PROTACs and self-assembling within cells, to provide a possible solution to this problem. Furthermore, lysosomal-based TPDs could induce the formation of endosomes and mediate the degradation of POIs via the lysosomal pathway, and TPDs do not necessarily enter the cell to function in the degradation of membrane-bound or circulating POIs. Incidentally, these TPDs usually contain monoclonal antibody structures to bind target proteins, so an intravenous administration must be taken into consideration.

Given its potential to degrade any target of choice, the reach of TPD can extend beyond oncology. As previously described, a key property of PROTACs is the ability to degrade proteins that are considered “undruggable” as they lack active sites. This attribute is especially engaging in targets for several neurodegenerative diseases involving the toxic build-up of proteins, such as tau, mutant huntingtin, etc. In addition, there are also efforts to develop TPDs for inflammation, immunity-associated diseases, and viral infection.

In conclusion, the field of TPD is full of challenges and opportunities and values both inspiration and effort. We also expect that after generations of development and validation, TPD will eventually provide new options for effective, safe, and reliable treatment for patients suffering from diseases.

## Figures and Tables

**Figure 1 pharmaceuticals-17-00100-f001:**
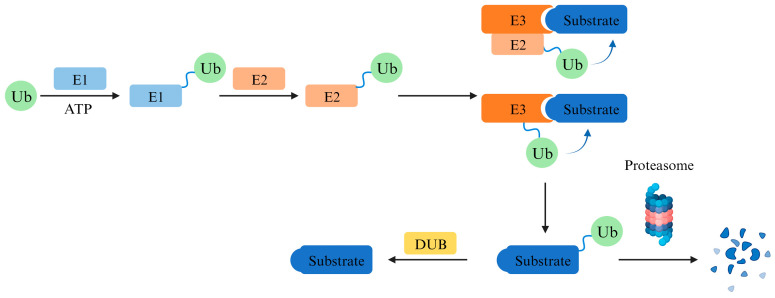
**Mechanism of ubiquitination.** E1 enzymes use ATP to generate a high-energy thioester between ubiquitin and the enzyme. Charged E1 enzymes transfer ubiquitin to E2 enzymes, which then cooperate with E3 ubiquitin ligases to produce a ubiquitylated substrate. E3 ubiquitin ligases can transfer ubiquitin directly from E2 enzymes or undergo charging of their reactive Cys from which ubiquitin is transferred to the substrate. The ubiquitylated substrate is then degraded by proteasome. This degradation event could also be terminated by DUBs cleaving ubiquitin off the substrate protein. Ub, ubiquitin; E1, ubiquitin-activating enzyme; E2, ubiquitin-conjugating enzyme; E3, ubiquitin ligase; DUB, deubiquitinating enzyme.

**Figure 2 pharmaceuticals-17-00100-f002:**
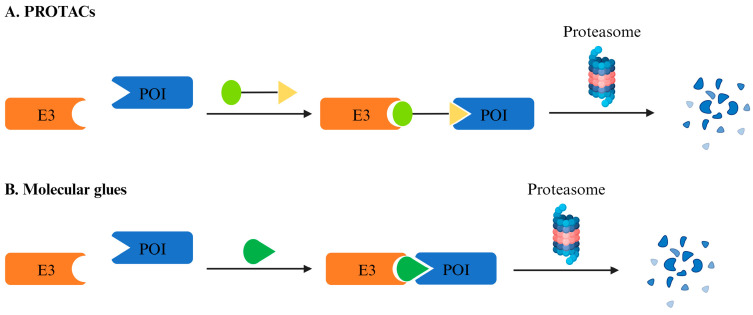
**Differences in the mechanism of action between PROTAC and molecular glue.** (**A**) PROTACs are composed of a target-binding warhead (green), a linker (black), and a ligase-binding moiety (yellow), the discovery of which usually undergoes a rational design. (**B**) The discovery of degrader molecular glues is usually serendipitous, whereby a known molecule is shown to have a degrader effect, making it possible to identify the E3 ligase mediating the degradation and determine whether the mechanism could be expanded to target additional POIs. E3, ubiquitin ligase; POI, protein of interest.

**Figure 3 pharmaceuticals-17-00100-f003:**
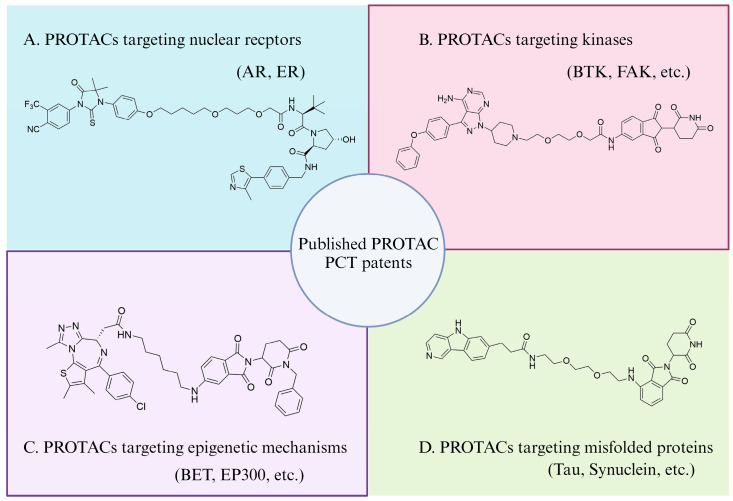
**A brief summarization of published PCT patents of PROTACs.** (**A**) PROTACs targeting nuclear receptors, exemplified by an AR-targeting PROTAC (patent number: WO2016118666A1). (**B**) PROTACs targeting kinases, exemplified by a BTK-targeting PROTAC (patent number: WO2019177902A1). (**C**) PROTACs targeting epigenetic mechanisms, exemplified by a BET-targeting PROTAC (patent number: WO2018064589A1). (**D**) PROTACs targeting misfolded proteins, exemplified by a Tau-targeting PROTAC (patent number: WO2019014429A1).

**Figure 4 pharmaceuticals-17-00100-f004:**
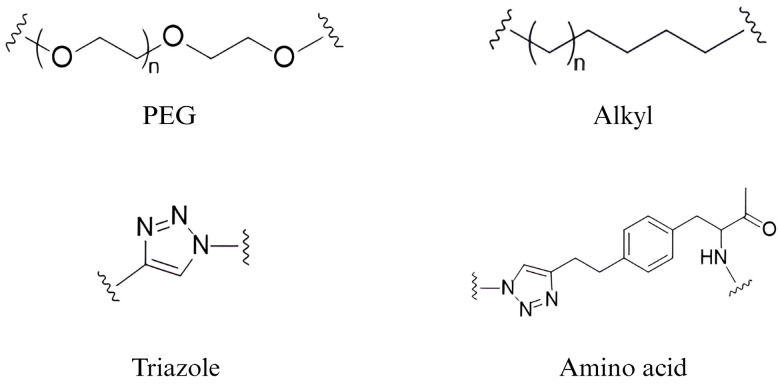
**Structure of some widely used PROTAC linkers.** PEG and alkyl chains are the most common linker motifs. Triazole groups are usually chosen to take advantage of click chemistry. Amino acid is a novel attempt to achieve the simultaneous targeting of two different POIs.

**Figure 5 pharmaceuticals-17-00100-f005:**
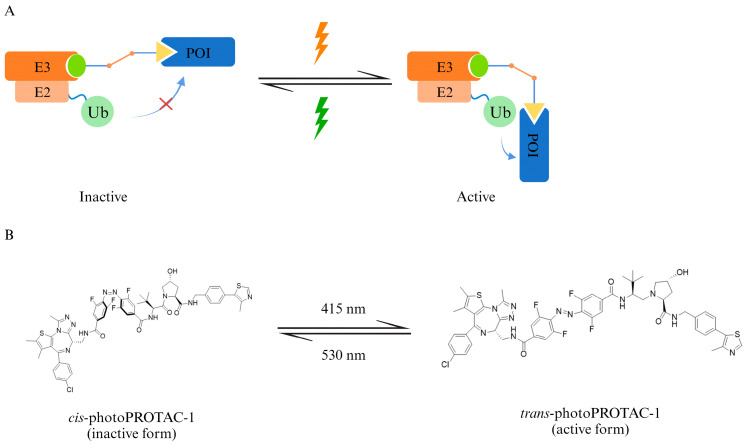
A photoswitchable PROTAC takes advantage of light-sensitive conformationally altered groups to achieve a reversible control of degradation. (**A**) Schematic diagram of photoswitchable PROTAC. (**B**) The light-dependent configuration change brings the inactive–active alteration of PROTAC. Ub, ubiquitin; E2, ubiquitin-conjugating enzyme; E3, ubiquitin ligase; POI, protein of interest.

**Figure 6 pharmaceuticals-17-00100-f006:**
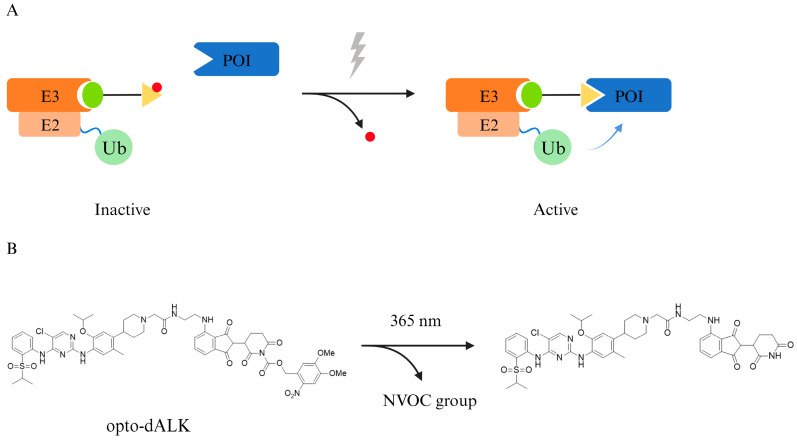
**A photocaged PROTAC uses light-unstable caging group to achieve a precise activation of protein degradation.** (**A**) Schematic diagram of photocaged PROTAC. (**B**) The NVOC group of opto-dALK is removed under a certain wavelength of UV radiation and therefore, the PROTAC molecule is fully activated. Ub, ubiquitin; E2, ubiquitin-conjugating enzyme; E3, ubiquitin ligase; POI, protein of interest.

**Figure 7 pharmaceuticals-17-00100-f007:**
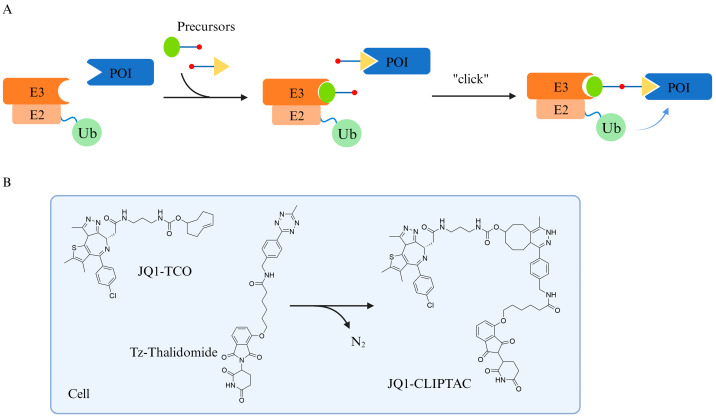
**CLIPTAC was proposed to improve the bioavailability of PROTAC.** (**A**) Schematic diagram of CLIPTAC principle. (**B**) The two precursors, JQ1-TCO and Tz-Thalidomide, can theoretically be self-assembled within the cell. Ub, ubiquitin; E2, ubiquitin-conjugating enzyme; E3, ubiquitin ligase; POI, protein of interest.

**Figure 8 pharmaceuticals-17-00100-f008:**
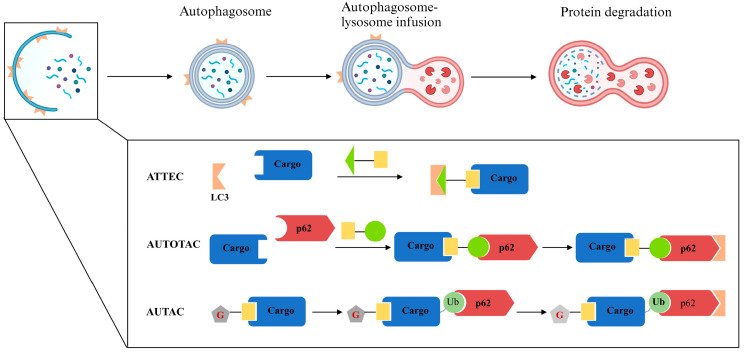
**TPD strategy with the help of autophagy.** During autophagy, a panel of autophagy-related gene products orchestrates the formation of an autophagosome, which encapsulates cellular cargo and fuses with lysosomes, resulting in the degradation of its contents. ATTEC mediates target cargo degradation by directly linking the cargo to LC3, an important marker protein on the phagosome surface. AUTOTAC takes advantage of p62, which mediates the transfer of ubiquitylated substrates into autophagic vesicles to degrade target cargo via the autophagy pathway. AUTAC attaches a cGMP-based autophagic degradation tag to the target cargo, leading to the degradation of certain POIs or organelles. Ub, ubiquitin; G, cGMP.

**Figure 9 pharmaceuticals-17-00100-f009:**
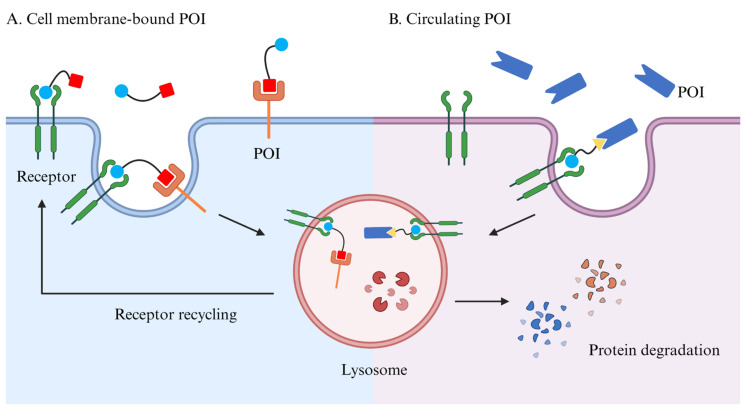
**Besides cytosolic POIs, lysosomal-based TPD enables the degradation of membrane-bound and extracellular proteins.** (**A**) Lysosomal-based TPD is capable of degrading cell-membrane-bound proteins without necessarily entering the cell. (**B**) Theoretically, lysosomal-based TPD can mediate the degradation of circulating POIs via the receptors located on the cell membrane. POI, protein of interest.

**Figure 10 pharmaceuticals-17-00100-f010:**
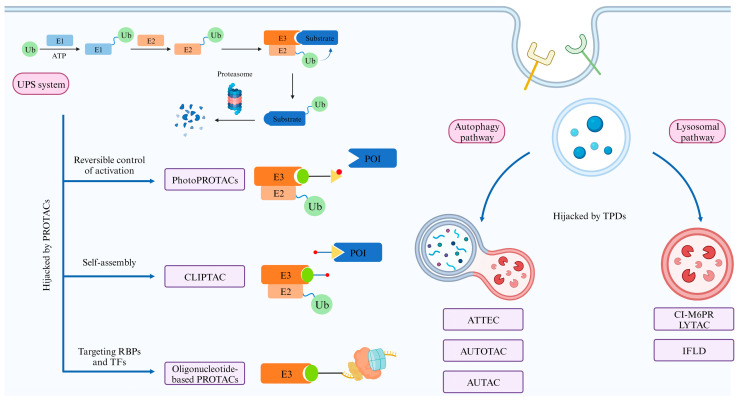
**An overview of the aforementioned explorations based on PROTACs and TPDs that take advantage of autophagy or lysosomal pathways.** PROTACs hijack the UPS system to degrade their POIs by directly binding both the E3 ligases and the target proteins. They could only degrade cytosolic proteins due to the intrinsic property of the UPS system. Several new explorations have been made to expand the application scenario. Moreover, some TPDs take advantage of degradation pathways apart from the UPS system. These autophagy- or lysosomal-pathway-based TPDs expand the target spectrum even further.

**Table 1 pharmaceuticals-17-00100-t001:** Other PROTACs that have entered clinical trials.

Company	Degrader	Target	States
Nurix Therapeutics (San Francisco, CA, USA)	NX-2127 [11]	BTK	Phase I (halted)
Nurix Therapeutics	NX-5948 [12]	BTK	Phase I (ongoing)
Kymera (Watertown, MA, USA)	KT-413 [13]	IKAR4	Phase I (ongoing)
Kymera	KT-333 [14]	STAT3	Phase I (ongoing)
C4 Therapeutics (Watertown, MA, USA)	CFT-8634	BRD9	Phase I/II, orphan drug designation
C4 Therapeutics	CFT-8919	EGFR-L858R	Phase I/Ⅱ
Dialectic Therapeutics (Dallas, TX, USA)	DT-2216	BCL-X_L_	Phase I, first track design
Foghorn Therapeutics (Cambridge, MA, USA)	FHD-609	BRD9	Phase I

## Data Availability

Data sharing is not applicable.

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
