# Peer review of "From PROTAC to TPD: Advances and Opportunities in Targeted Protein Degradation"

_pharmaceuticals, 2024, doi:10.3390/ph17010100_

Round 1

Reviewer 1 Report

Comments and Suggestions for Authors

PROTAC or PROteolysis TAgeting Chimeras, is a therapeutic approach in drug development that aims to degrade target proteins within cells. Traditional drug development often focuses on inhibiting the function of specific proteins, but PROTACs take a different approach by promoting the degradation of these proteins. Design small PROTACs to bind simultaneously to the target protein of interest and an E3 ubiquitin ligase, allowing ubiquitination of the target protein, and degradation. However, this therapeutic modality draws attention due to its ability to target undruggable proteins.

As the field progresses, a comprehensive understanding of PROTAC design and targeting different proteins of interest using cellular degradation machinery are vital. Hence, the scientific aspect of the current review article is valuable and on demand. Here, the authors summarized the key findings through text mining and referencing. Also, important diagrams make the manuscript more informative.

However, the manuscript lacks a crucial aspect in the conclusion and needs further improvement. Here, I offer some suggestions.

1) Challenges in developing PROTAC, large size molecules, less number of available E3 ligases that can be used to target substrates, off-target effect, etc. are not discussed briefly by the authors. Hence, the authors are suggested to include these aspects in the conclusion section.

2) Authors should consider adding the wavelength of light used in PhotoPROTAC in Figure 4, to be more informative. Same for the figure 5.

3) Labeling of the figure 7 is important. Readers from different backgrounds might not be able to follow it, so authors should describe the figure. Also, labeling the figures and adding small descriptions in all the figures (Fig. 1- Fig. 8) will help the readers to follow and understand better. The authors can also consider integrating different approaches into one or two figures.

4) Authors should elaborate on the lysosomal pathway in the figure 8. Very few molecular details were demonstrated.

Author Response

Thank you for your valuable advice. Please see the attachment for detailed responses.

Reviewer 2 Report

Comments and Suggestions for Authors

I have gone through the entire review article. The writing part was very impressive. The authors have written a good introduction. Maintained very good flow in the entire research article. In my opinion, the submitted review paper should go through minor revision. The comments are mentioned below, please address them accordingly.

1)Please do describe PROTAC patents.

2)Diagrams were not clear, please provide quality images. 

3)Kindly include future directions for PROTACs

Author Response

(The authors gave the same response as above.)

Reviewer 3 Report

Comments and Suggestions for Authors

Various targeted protein degradation (TPD) strategies have been well presented in the last decade, where different proteins were targetted through this approach. 

The authors attempted to compile the recent development of TPD and associated strategies. There are a few sections that need further elaboration.

Point 1. Light-acting PROTACs, or Photo-PROTACs or Azo-Protacs

This strategy is relatively recent and has shown a promising application where, at a specific wavelength, when irradiated, the PROTAC gets activated to its binding configuration (Trans-form more often in its binding configuration) suitable to bind to targetted proteins (or protein of interest). Therefore, this strategy is organized into Photoswitchable PROTACs and, to an extent Photocaged-PROTACs (Photocaged-PROTACs, are those PROTACs, when irradiated irreversible release a photolabile group and become active to bind to their protein of interest). In its current form, the manuscript requires more elaboration on this strategy. However, it is also not possible to cover all the aspects of the strategy; therefore, I would like to ask the author to  compile (at least) the important examples of Photocaged and Photoswitcale PROTACs

Here are some references that would help authors, or authors can choose examples.

1. 10.1126/sciadv.aay506

2.doi.org/10.1002/slct.202200981

3.10.1021/acscentsci.9b00713

Point 2: The author should also put forward their expert opinion on the limitations of these PROTAC strategies as there are various case-to-case scenarios where one strategy is more applicable than strategy. Please elaborate further.

Author Response

(The authors gave the same response as above.)
